# Cinematic Mindscapes: High-quality Video Reconstruction from Brain Activity

**Zijiao Chen**[*]
National University of Singapore
`zijiao.chen@u.nus.edu`

**Jiaxin Qing**[*]
The Chinese University of Hong Kong
`jqing@ie.cuhk.edu.hk`

**Juan Helen Zhou**[†]
National University of Singapore
`helen.zhou@nus.edu.sg`
*https://mind-video.com*

## Abstract

Reconstructing human vision from brain activities has been an appealing task that helps to understand our cognitive process. Even though recent research has seen great success in reconstructing static images from non-invasive brain recordings, work on recovering continuous visual experiences in the form of videos is limited. In this work, we propose **MinD-Video** that learns spatiotemporal information from continuous fMRI data of the cerebral cortex progressively through masked brain modeling, multimodal contrastive learning with spatiotemporal attention, and co-training with an augmented Stable Diffusion model that incorporates network temporal inflation. We show that high-quality videos of arbitrary frame rates can be reconstructed with **MinD-Video** using adversarial guidance. The recovered videos were evaluated with various semantic and pixel-level metrics. We achieved an average accuracy of 85% in semantic classification tasks and 0.19 in structural similarity index (SSIM), outperforming the previous state-of-the-art by 45%. We also show that our model is biologically plausible and interpretable, reflecting established physiological processes.

## 1 Introduction

Life unfolds like a film reel, each moment seamlessly transitioning into the next, forming a "perpetual theater" of experiences. This dynamic narrative forms our perception, explored through the naturalistic paradigm, painting the brain as a moviegoer engrossed in the relentless film of experience. Understanding the information hidden within our complex brain activities is a big puzzle in cognitive neuroscience. The task of recreating human vision from brain recordings, especially using non-invasive tools like functional Magnetic Resonance Imaging (fMRI), is an exciting but difficult task. Non-invasive methods, while less intrusive, capture limited information, susceptible to various interferences like noise [1]. Furthermore, the acquisition of neuroimaging data is a complex, costly process. Despite these complexities, progress has been made, notably in learning valuable fMRI features with limited fMRI-annotation pairs. Deep learning and representation learning have achieved significant results in visual class detections [2, 3] and static image reconstruction [4, 5, 6, 7, 8], advancing our understanding of the vibrant, ever-changing spectacle of human perception.

Unlike still images, our vision is a continuous, diverse flow of scenes, motions, and objects. To recover dynamic visual experience, the challenge lies in the nature of fMRI, which measures blood oxygenation level dependent (BOLD) signals and captures snapshots of brain activity every few seconds. Each fMRI scan

---
[*]Equal contributions
[†]Corresponding author

37th Conference on Neural Information Processing Systems (NeurIPS 2023).

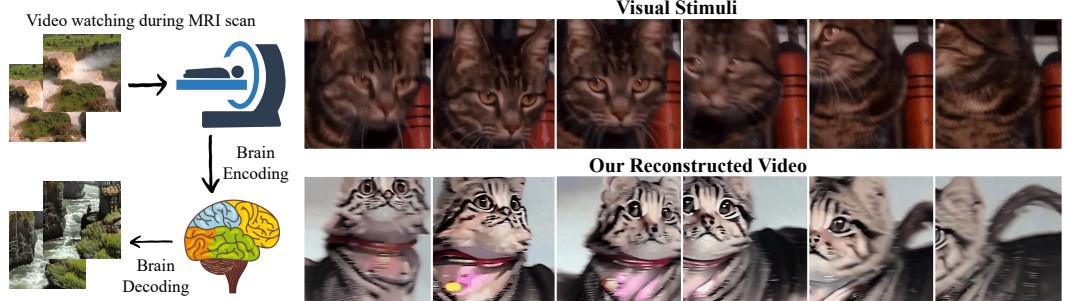

Figure 1. **Brain decoding & video reconstruction**. We propose a progressive learning approach to recover continuous visual experience from fMRI. High-quality videos with accurate semantics and motions are reconstructed.

essentially represents an "average" of brain activity during the snapshot. In contrast, a typical video has about 30 frames per second (FPS). If an fMRI frame takes 2 seconds, during that time, 60 video frames - potentially containing various objects, motions, and scene changes - are presented as visual stimuli. Thus, decoding fMRI and recovering videos at an FPS much higher than the fMRI's temporal resolution is a complex task.

Hemodynamic response (HR) [9] refers to the lags between neuronal events and activation in BOLD signals. When a visual stimulus is presented, the recorded BOLD signal will have certain delays with respect to the stimulus event. Moreover, the HR varies across subjects and brain regions [10]. Thus, the common practice that shifts the fMRI by a fixed number in time to compensate for the HR would be sub-optimal.

In this work, we present **MinD-Video**, a two-module pipeline designed to bridge the gap between image and video brain decoding. Our model progressively learns from brain signals, gaining a deeper understanding of the semantic space through multiple stages in the first module. Initially, we leverage large-scale unsupervised learning with masked brain modeling to learn general visual fMRI features. We then distill semantic-related features using the multimodality of the annotated dataset, training the fMRI encoder in the Contrastive Language-Image Pre-Training (CLIP) space with contrastive learning. In the second module, the learned features are fine-tuned through co-training with an augmented stable diffusion model, which is specifically tailored for video generation under fMRI guidance. Our contributions are summarized as follows:

- We introduced a flexible and adaptable brain decoding pipeline decoupled into two modules: an fMRI encoder and an augmented stable diffusion model, trained separately and finetuned together.
- We designed a progressive learning scheme where the encoder learns brain features through multiple stages, including multimodal contrastive learning with spatiotemporal attention for windowed fMRI.
- We augmented the stable diffusion model for scene-dynamic video generation with near-frame attention. We also designed adversarial guidance for distinguishable fMRI conditioning.
- We recovered high-quality videos with accurate semantics, e.g., motions and scene dynamics. Results are evaluated with semantic and pixel metrics at video and frame levels. An accuracy of 85% is achieved in semantic metrics and 0.19 in SSIM, outperforming the previous state-of-the-art approaches by 45%.
- The attention analysis revealed mapping to the visual cortex and higher cognitive networks, suggesting our model is biologically plausible and interpretable.

## 2   Background

**Image Reconstruction** Image reconstruction from fMRI was first explored in [2], which showed that hierarchical image features and semantic classes could be decoded from the fMRI data collected when the participants were looking at a static visual stimulus. Authors in [5, 6] designed a separable autoencoder that enables self-supervised learning in fMRI and images to increase training data. Based on a similar philosophy, [7] proposed to perform self-supervised learning on a large-scale fMRI dataset using masked data modeling as a pretext task. Using a stable diffusion model as a generative prior and the pre-trained fMRI features as conditions, [7] reconstructed high-fidelity images with high semantic correspondence to the groundtruth stimuli.

**Video Reconstruction** The conventional method formulated the video reconstruction as multiple image reconstructions [11], leading to low frame rates and frame inconsistency. Nonetheless, [11] showed that low-level image features and classes can also be decoded from fMRI collected with video stimulus. Using

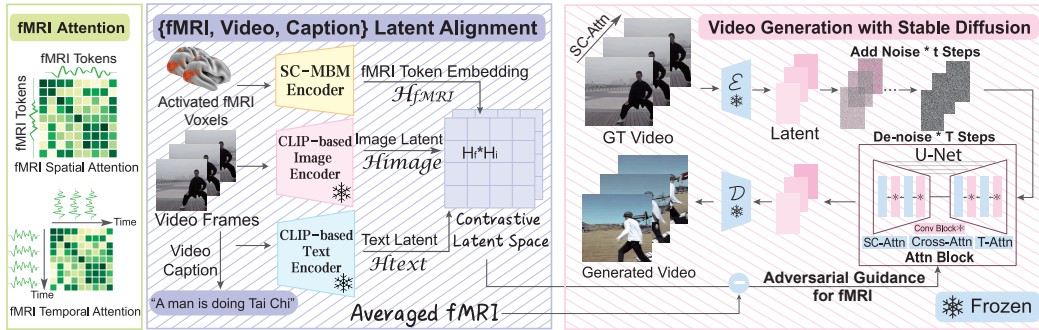

Figure 2. **MinD-Video Overview**. Our method has two modules that are trained separately and then finetuned together. The fMRI encoder progressively learns fMRI features through multiple stages, including MBM pre-training and multimodal contrastive learning. A spatiotemporal attention is designed to process multiple fMRI in a sliding window. The augmented Stable Diffusion is trained with videos and then tuned with the fMRI encoder using annotated data.

fMRI representations encoded with a linear layer as conditions, [12] generated higher quality and frame rate videos with a conditional video GAN. However, the results are limited by data scarcity, especially for GAN training, which generally requires a large amount of data. [13] took a similar approach as [5], which relied on the same separable autoencoder that enables self-supervised learning. Even though better results were achieved than [12], the generated videos were of low visual fidelity and semantic meanings. In addition, recent work by [14] utilizes contrastive learning to derive latent embeddings from mice neural data, enabling the generation of videos based on these learned representations.

**MBM** Masked brain modeling (MBM) is a pre-text task that enables self-supervised learning in a large fMRI dataset proposed in [7], aiming to build a brain foundation model. It learns general features of fMRI by trying to recover masked data from the remainings, similar to the GPT [15] and MAE [16], after which knowledge can be distilled and transferred to a downstream task with limited data and a few-step tuning [17, 15, 16, 7].

**CLIP** Contrastive Language-Image Pre-Training (CLIP) is a pre-training technique that builds a shared latent space for images and natural languages by large-scale contrastive learning [18]. The training aims to minimize the cosine distance of paired image and text latent while maximizing permutations of pairs within a batch. The shared latent space (CLIP space) contains rich semantic information on both images and texts.

**Stable Diffusion** Diffusion models are emerging probabilistic generative models defined by a reversible Markov chain [19, 20]. As a variant, stable diffusion generates a compressed version of the data (data latent) instead of generating the data directly. As it works in the data latent space, the computational requirement is significantly reduced, and higher-quality images with more details can be generated in the latent space [21]. Due to its high generative quality, it has been studied in various brain decoding tasks [7, 8].

## 3 Methodology

### 3.1 Motivation and Overview

Aiming for a flexible design, MinD-Video is decoupled into two modules: an fMRI encoder and a video generative model, as illustrated in Fig. 2. These two modules are trained separately and then finetuned together, which allows for easy adaption of new models if better architectures of either one are available. As a representation learning model, the encoder in the first module transfers the pre-processed fMRI into embeddings, which are used as a condition for video generations. For this purpose, the embedding should have the following traits: 1) It should contain rich and compact information about the visual stimulus presented during the scan. 2) It should be close to the embedding domain with which the generative model is trained. When designing the generative model, it is essential that the model produces not only diverse, high-quality videos with high computational efficiency but also handles potential scene transitions, mirroring the dynamic visual stimuli experienced during scans.

## 3.2 The fMRI Pre-processing

The fMRI captures whole-brain activity with BOLD signals (voxels). Each voxel is assigned to a region of interest (ROI) for focused analysis. Here, we concentrate on voxels activated during visual stimuli. There are two ways to define the ROIs: one uses a pre-defined parcellation such as [22] and [23] to obtain the visual cortex; the other relies on statistical tests to identify activated voxels during stimuli. Our large-scale pre-training is based on the parcellation in [22], while statistical tests are employed for the target dataset, i.e., the Wen (2018) [11] dataset containing fMRI-video pairs. To determine activated regions, we calculate intra-subject reproducibility of each voxel, correlating fMRI data across multiple viewings. The correlation coefficients are then converted to z-scores and averaged. We compute statistical significance using a one-sample t-test (P<0.01, DOF=17, Bonferroni correction). We selected the top 50% of the most significant voxels after the statistical test. Notice that most of the identified voxels are from the visual cortex.

## 3.3 Progressive Learning and fMRI Encoder

Progressive learning is used as an efficient training scheme where general knowledge is learned first, and then more task-specific knowledge is distilled through finetuning [24, 25, 26]. To generate meaningful embeddings specific for visual decoding, we design a progressive learning pipeline, which learns fMRI features through multiple stages, starting from general features to more specific and semantic-related features. We will show that the progressive learning process is reflected biologically in the evolution of fMRI attention maps.

**Large-scale Pre-training with MBM** Similar to [7], a large-scale pre-training with masked brain modeling (MBM) is used to learn general features of the visual cortex. With the same setup, an asymmetric vision-transformer-based autoencoder [27, 16] is trained on the Human Connectome Project [28] with the visual cortex (V1 to V4) defined by [22]. Specifically, fMRI data of the visual cortex is rearranged from 3D into 1D space in the order of visual processing hierarchy, which is then divided into patches of the same size. The patches will be transformed into tokens, and a large portion ($\sim$75%) of the tokens will be randomly masked in the encoder during training. With the autoencoder architecture, a simple decoder aims to recover the masked tokens from the unmasked token embeddings generated by the encoder. The main idea behind the MBM is that if the training objective can be achieved with high accuracy using a simple decoder, the token embeddings generated by the encoder will be a rich and compact description of the original fMRI data. We refer readers to [7] for detailed descriptions and reasonings of the MBM.

**Spatiotemporal Attention for Windowed fMRI** For the purpose of image reconstruction, the original fMRI encoder in [7] shifts the fMRI data by 6s, which is then averaged every 9 seconds and processed individually. This process only considers the spatial information in its attention layers. In the video reconstruction, if we directly map one fMRI to the video frames presented (e.g., 6 frames), the video reconstruction task can be formulated as a one-to-one decoding task, where each set of fMRI data corresponds to 6 frames. We call each {fMRI-frames} pair a *fMRI frame window*. However, this direct mapping is sub-optimal because of the time delay between brain activity and the associated BOLD signals in fMRI data due to the nature of hemodynamic response.

Thus, when a visual stimulus (i.e., a video frame) is presented at time $t$, the fMRI data obtained at $t$ may not contain complete information about this frame. Namely, a lag occurs between the presented visual stimulus and the underlying information recorded by fMRI. This phenomenon is depicted in Fig. 3.

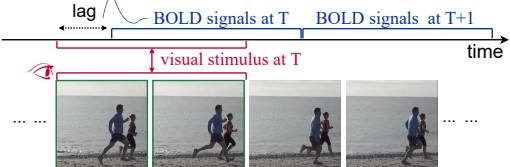

Figure 3. Due to hemodynamic response, the BOLD signal (blue) lags a few seconds behind the visual stimulus (red), causing a discrepancy between fMRI and the stimulus.

The hemodynamic response function (HRF) is usually used to model the relationship between neural activity and BOLD signals [29]. In an LTI system, the signal $y(t)$ is represented as the convolution of a stimulus function $s(t)$ and the HR $h(t)$, i.e., $y(t) = (s * h)(t)$. The $h(t)$ is often modeled with a linear combination of some basis functions, which can be collated into a matrix form: $Y = X\beta + e$, where $Y$ represents the observed data, $\beta$ is a vector of regression coefficients, and $e$ is a vector of unexplained error values. However, $e$ varies significantly across individuals and sessions due to age, cognitive state, and specific visual stimuli, which influence the firing rate, onset latency, and neuronal activity duration. These variations impact the estimation of $e$ and ultimately affect the accuracy of the fMRI-based analysis. Gener-

ally, there are two ways to address individual variations: using personalized HRF models or developing algorithms that adapt to each participant. We choose the latter due to its superior flexibility and robustness.

Aiming to obtain sufficient information to decode each scan window and account for the HR, we propose a spatiotemporal attention layer to process multiple fMRI frames in a sliding window. Consider a sliding window defined as $\boldsymbol{x_t} = \{x_t, x_{t+1}, ..., x_{t+w-1}\}$, where $x_t \in \mathbb{R}^{n \times p \times b}$ are token embeddings of the fMRI at $t$ and $n, p, b$ are the batch size, patch size, and embedding dimension, respectively. So we have $\boldsymbol{x_t} \in \mathbb{R}^{n \times w \times p \times b}$, where $w$ is the window size. Recall that the attention is given by attn=softmax$\left(\frac{QK^T}{\sqrt{d}}\right)$. To calculate spatial attention, we use the network inflation trick [30], where we merge the first two dimensions of $\boldsymbol{x_t}$ and obtain $\boldsymbol{x_t^{spat}} \in \mathbb{R}^{nw \times p \times b}$. Then the query and key are calculated in Eq. (1) as

$$Q = \boldsymbol{x_t^{spat}} \cdot W_{spat}^Q, \quad K = \boldsymbol{x_t^{spat}} \cdot W_{spat}^K. \tag{1}$$

Likewise, we merge the first and the third dimension of $\boldsymbol{x_t}$ to calculate the temporal attention, obtaining $\boldsymbol{x_t^{temp}} \in \mathbb{R}^{np \times w \times b}$. Again, the query and key are calculated in Eq. (2) with

$$Q = \boldsymbol{x_t^{temp}} \cdot W_{temp}^Q, \quad K = \boldsymbol{x_t^{temp}} \cdot W_{temp}^K. \tag{2}$$

The spatial attention learns correlations among the fMRI tokens, describing the spatial correlations of the fMRI patches. Then the temporal attention learns the correlations of fMRI from the sliding window, including sufficient information to cover the lag due to HR, as illustrated in the "fMRI Attention" in Fig. 2.

**Multimodal Contrastive Learning** Recall that the fMRI encoder is pre-trained to learn general features of the visual cortex, and then it is augmented with temporal attention heads to process a sliding window of fMRI. In this step, we further train the augmented encoder with {fMRI, image, caption} triplets and pull the fMRI embeddings closer to a shared CLIP space containing rich semantic information. Additionally, the generative model is usually pre-trained with text conditioning. Thus, pulling the fMRI embeddings closer to the text-image shared space ensures its understandability by the generative model during conditioning.

Firstly, videos in the training set are downsampled to a smaller frame rate (3FPS). Each frame is then captioned with BLIP [31], which creates the {image, text} pairs. With the CLIP text encoder and image encoder being fixed, the CLIP loss [18] is calculated for fMRI-image and fMRI-text, respectively. Denote the pooled text embedding, image embedding, and fMRI embedding by $emb_t, emb_i, emb_f \in \mathbb{R}^{n \times b}$. The contrastive language-image-fMRI loss is given by

$$\mathcal{L} = (\mathcal{L}_{\text{CLIP}}(emb_f, emb_t) + \mathcal{L}_{\text{CLIP}}(emb_f, emb_i))/2, \tag{3}$$

where $\mathcal{L}_{\text{CLIP}}(a, b) = \text{CrossEntropy}(\epsilon a \cdot b^{\text{T}}, [0,1,...,n])$, with $\epsilon$ being a scaling factor. Extra care is needed to reduce similar frames in a batch for better contrastive pairs. From Eq. (3), we see that the loss largely depends on the batch size $n$. Thus, a large $n$ with data augmentation on all modalities is appreciated.

## 3.4 Video Generative Module

The Stable Diffusion model [21] is used as the base generative model considering its excellence in generation quality, computational requirements, and weights availability. However, as the stable diffusion model is an image-generative model, temporal constraints need to be applied in order for video generation.

**Scene-Dynamic Sparse Causal (SC) Attention** Authors in [30] use a network inflation trick with sparse temporal attention to adapt the stable diffusion to a video generative model. Specifically, the sparse temporal attention effectively conditions each frame on its previous frame and the first frame, which ensures frame consistency and also keeps the scene unchanged. However, the human vision consists of possible scene changes, so the video generation should also be scene-dynamic. Thus, we relax the constraint in [30] and condition each frame on its previous two frames, ensuring the video smoothness while allowing scene dynamics. Using notations from [30], the SC attention is calculated with the query, key, and value given by

$$Q = W^Q \cdot z_{v_i}, \quad K = W^K \cdot [z_{v_{i-2}}, z_{v_{i-1}}], \quad V = W^V \cdot [z_{v_{i-2}}, z_{v_{i-1}}], \tag{4}$$

where $z_{v_i}$ denotes the latent of the $i$-th frame during the generation.

**Adversarial Guidance for fMRI** Classifier-free guidance is widely used in the conditional sampling of diffusion models for its flexibility and generation diversity, where the noise update function is given by

$$\hat{\epsilon}_\theta(z_t, c) = \epsilon_\theta(z_t) + s(\epsilon_\theta(z_t, c) - \epsilon_\theta(z_t)), \tag{5}$$

where $c$ is the condition, $s$ is the guidance scale and $\epsilon_\theta(\cdot)$ is a score estimator implemented with UNet [32]. Interestingly, Eq. (5) can be changed to an adversarial guidance version [21]

$$\hat{\epsilon}_\theta(z_t, c, \bar{c}) = \epsilon_\theta(z_t, \bar{c}) + s(\epsilon_\theta(z_t, c) - \epsilon_\theta(z_t, \bar{c})), \tag{6}$$

where $\bar{c}$ is the negative guidance. In effect, generated contents can be controlled through "what to generate" (positive guidance) and "what not to generate" (negative guidance). When $\bar{c}$ is a null condition, the noise update function falls back to Eq. (5), the regular classifier-free guidance. In order to generate diverse videos for different fMRI, guaranteeing the distinguishability of the inputs, we average all fMRI in the testing set and use the averaged one as the negative condition. Specifically, for each fMRI input, the fMRI encoder will generate an unpooled token embedding $x \in \mathbb{R}^{l \times b}$, where $l$ is the latent channel number. Denote the averaged fMRI as $\bar{x}$. We have the noise update function $\hat{\epsilon}_\theta(z_t, x, \bar{x}) = \epsilon_\theta(z_t, \bar{x}) + s(\epsilon_\theta(z_t, x) - \epsilon_\theta(z_t, \bar{x}))$.

**Divide and Refine** With a decoupled structure, two modules are trained separately: the **fMRI encoder** is trained in a large-scale dataset and then tuned in the target dataset with contrastive learning; the **generative module** is trained with videos and captions from the target dataset. In the second phase, two modules are tuned together with fMRI-video pairs, where the encoder and part of the generative model are trained. Different from [30], we tune the whole self-attention, cross-attention, and temporal-attention heads instead of only the query projectors, as a different modality is used for conditioning. The second phase is also the last stage of encoder progressive learning, after which the encoder finally generates token embeddings that contain rich semantic information and are easy to understand by the generative model.

### 3.5    Learning from the Brain - Interpretability

Our objectives extend beyond brain decoding and reconstruction. We also aim to understand the biological principles of the decoding process. To this end, we visualize average attention maps from the first, middle, and last layers of the fMRI encoder across all testing samples. This approach allows us to observe the transition from capturing local relations in early layers to recognizing more global, abstract features in deeper layers [33]. Additionally, attention maps are visualized for models in different learning stages: large-scale pre-training, contrastive learning, and co-training. By projecting attention back to brain surface maps, we can easily visualize each brain region's contributions and the learning progress through each stage.

## 4    Experiments

### 4.1    Datasets

**Pre-training dataset** Human Connectome Project (HCP) 1200 Subject Release [28]: For our upstream pre-training dataset, we employed resting-state and task-evoked fMRI data from the HCP. Building upon [7], we obtained 600,000 fMRI segments from a substantial amount of fMRI scan data.

**Paired fMRI-Video dataset** A publicly available benchmark fMRI-video dataset [11] was used, comprising fMRI and video clips. The fMRI were collected using a 3T MRI scanner at a TR of 2 seconds with three subjects. The training data included 18 segments of 8-minute video clips, totaling 2.4 video hours and yielding 4,320 paired training examples. The test data comprised 5 segments of 8-minute video clips, resulting in 40 minutes of test video and 1,200 test fMRIs. The video stimuli were diverse, covering animals, humans, and natural scenery, and featured varying lengths at a temporal resolution of 30 FPS.

### 4.2    Implementation Details

The original videos are downsampled from 30 FPS to 3 FPS for efficient training and testing, leading to 6 frames per fMRI frame. In our implementation, we reconstruct a video of 2 seconds (6 frames) from a sliding window of fMRI frames. However, thanks to the spatiotemporal attention head design that encodes multiple fMRI at once, our method can reconstruct longer videos from more fMRI frames if more GPU memory is available.

A ViT-based fMRI encoder with a patch size of 16, a depth of 24, and an embedding dimension of 1024 is used. After pre-training with a mask ratio of 0.75, the encoder will be augmented with a projection head that projects the token embedding into the dimension of $77 \times 768$. The Stable Diffusion V1-5 [21] trained at the resolution of $512 \times 512$ is used. But we tune the augmented Stable Diffusion for video generations at the resolution of $256 \times 256$ with 3 FPS. Notice that the FPS and image are downsampled for efficient

experiments, and our method can also work with full temporal and spatial resolution. All parameters in the fMRI encoder pre-training are the same as [7] with eight RTX3090, while other stages are trained with one RTX3090.

The fMRI encoder pre-training on a large-scale dataset is the most resource-consuming part, but only one pre-training is needed for different subjects. This is in contrast to the other downstream stages. Multimodal contrastive learning and co-training are performed for each subject individually due to individual variability. The inference is performed with 200 DDIM [34] steps. See Supplementary for more details.

### 4.3  Evaluation Metrics

Following prior studies, we utilize both frame-based and video-based metrics. Frame-based metrics evaluate each frame individually, providing a snapshot evaluation, whereas video-based metrics assess sequences of frames, encapsulating the dynamics across frames. Both are used for a comprehensive analysis. Unless stated otherwise, all test set videos are used for evaluating the three subjects.

**Frame-based Metrics** Our frame-based metrics are divided into two classes, pixel-level metrics and semantics-level metrics. We use the structural similarity index measure (SSIM) [35] as the pixel-level metric and the N-way top-K accuracy classification test as the semantics-level metric. Specifically, for each frame in a scan window, we calculate the SSIM and classification test accuracy with respect to the groundtruth frame. To perform the classification test, we basically compare the classification results of the groundtruth (GT) and the predicted frame (PF) using an ImageNet classifier. If the GT class[3] is within the top-K probability of the PF classification results from N randomly picked classes, including the GT class, we declare a successful trial. The test is repeated for 100 times, and the average success rate is reported.

**Video-based Metric** The video-based metric measures the video semantics using the classification test as well, except that a video classifier is used. The video classifier based on VideoMAE [36] is trained on Kinetics-400 [37], an annotated video dataset with 400 classes, including motions, human interactions, etc.

## 5  Results

We compare our method against three fMRI-video baselines: Wen et al. (2018) [11], Wang et al. (2022) [12] and Kupershmidt et al. (2022) [13]. Visual comparisons are shown in Fig. 4, and quantitative comparisons are shown in Fig. 6, where publicly available data and samples are used for comparison. As shown, we generate high-quality videos with more semantically meaningful content. Following the literature, we evaluate the SSIM of Subject 1 with the first testing video, achieving a score of 0.186, outperforming the state-of-the-art by 45%. When comparing with Kupershmidt et al. (2022), we evaluate all test videos for different subjects, and our method outperforms by 35% on average, as shown in Fig. 6. Using semantic-level metrics, our method achieves a success rate of 0.849 and 0.2 in 2-way and 50-way top-1 accuracy classification tests, respectively, with the video classifier. The image classifier gives a success rate of 0.795 and 0.19, respectively, with the same tests, which significantly surpasses the chance level of these two tests (2-way: 0.5, 50-way: 0.02). Full results are shown in Tab. 1. We also compare our results with an image-fMRI model by Chen et al. (2023) [7]. An image is produced for each fMRI, samples of which are shown in Fig. 4 with the "walking man" as the groundtruth. Even though the results and groundtruth are semantically matching, frame consistency and image quality are not satisfying. A lag due to the hemodynamic response is also observed. The first frame actually corresponds to the previous groundtruth.

**Different Subjects** After fMRI pre-processing, each subject varies in the size of ROIs, where Subject 1, 2 and 3 have 6016, 6224, and 3744 voxels, respectively. As Subject 3 has only half the voxels of the others, a larger batch size can be used during contrastive learning, which may lead to better results as shown in Tab. 1. Nonetheless, all subjects are consistent in both numeric and visual evaluations (See Supplementary).

**Recovered Semantics** In the video reconstruction, we define the semantics as the objects, animals, persons, and scenes in the videos, as well as the motions and scene dynamics, e.g., people running, fast-moving scenes, close-up scenes, long-shot scenes, etc. We show that even though the fMRI has a low temporal resolution, it contains enough information to recover the mentioned semantics. Fig. 5 shows a few examples of reconstructed frames using our method. Firstly, we can see that the basic objects, animals, persons, and scene types can be well recovered. More importantly, the motions, such as running, dancing, and singing,

---

[3]Since video frames usually cannot be well described with a single class, we use the top-3 classification results as the GT class, and a successful event is declared if the test succeeds with any of the GT class.

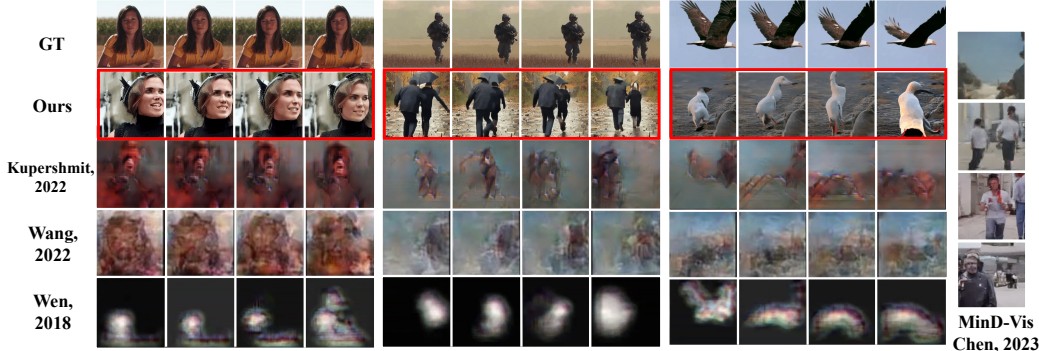

Figure 4. **Compare with Benchmarks**. We compare our results with the samples provided in the previous literature. Our method generates samples that are more semantically meaningful and match with the groundtruth.

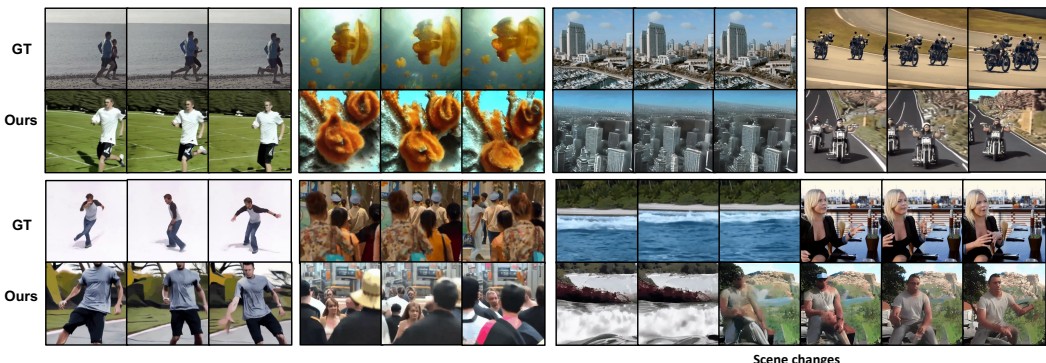

Figure 5. **Reconstruction Diversity**. Various motions, scenes, persons, and animals can be correctly recovered. A sample with a scene transition is shown on the bottom right.

Table 1. **Ablation study** on window sizes, multimodal contrastive learning, and adversarial guidance (AG). Evaluations on different subjects are also shown. Full Model: win=2, Sub 1. Colors reflect statistical significance (two-sample t-test) compared to the Full Model. $p < 0.0001$ (purple); $p < 0.01$ (pink); $p < 0.05$ (yellow); $p > 0.05$ (green)

| | Video-based | | Frame-based | | |
| --- | --- | --- | --- | --- | --- |
| | Semantic-level | | Semantic-level | | Pixel-level |
| | 2-way↑ | 50-way↑ | 2-way↑ | 50-way↑ | SSIM↑ |
| **Full Model** | **0.853**±0.03 | **0.202**±0.02 | **0.792**±0.03 | **0.172**±0.01 | **0.171** |
| Window Size = 1 | 0.851±0.03 | 0.195±0.02 | 0.759±0.03 | 0.165±0.01 | 0.169 |
| Window Size = 3 | 0.826±0.03 | 0.161±0.01 | 0.765±0.03 | 0.137±0.01 | 0.161 |
| w/o Contrastive | 0.844±0.03 | 0.157±0.02 | 0.750±0.03 | 0.088±0.07 | 0.135 |
| Text-fMRI Contra | 0.839±0.03 | 0.185±0.01 | 0.78±0.03 | 0.154±0.01 | 0.164 |
| Img-fMRI Contra | 0.845±0.03 | 0.189±0.01 | 0.783±0.03 | 0.151±0.01 | 0.164 |
| w/o AG | 0.859±0.03 | 0.198±0.02 | 0.775±0.03 | 0.117±0.01 | 0.152 |
| Subject 2 | 0.841±0.03 | 0.173±0.02 | 0.784±0.03 | 0.158±0.13 | 0.171 |
| Subject 3 | 0.846±0.03 | 0.216±0.02 | 0.812±0.03 | 0.193±0.01 | 0.187 |

and the scene dynamics, such as the close-up of a person, the fast-motion scenes, and the long-shot scene of a city view, can also be reconstructed correctly. This result is also reflected in our numerical metrics, which consider both frame semantics and video semantics, including various categories of motions and scenes.

**Abalations** We test our method using different window sizes, starting from a window size of 1 up to 3. When the window size is one, the fMRI encoder falls back to a normal MBM encoder in [7]. Tab. 1 shows that when all other parameters are fixed, a window size of 2 gives the best performance in general, which is reasonable as the hemodynamic response usually will not be longer than two scan windows. Additionally, we also test the effectiveness of multimodal contrastive learning. As shown in Tab. 1, without contrastive

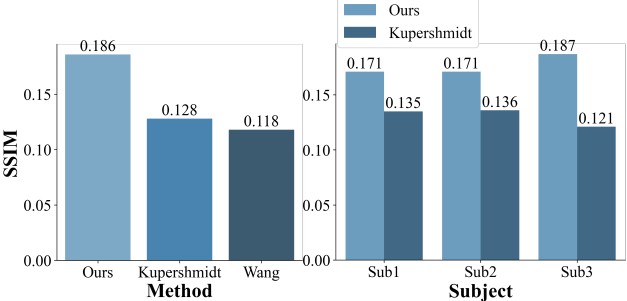

Figure 6. **SSIM Comparision**. Left: comparison with Kupershmidt et al. (2022) [13] and Wang et al. (2022) [12] on commonly available test samples (Subject 1, test video 1). Right: comparison with Kupershmidt et al. (2022) on all subjects, all test videos.

learning, the generation quality degrades significantly. When two modalities are used, either text-fMRI or image-fMRI, the performance is inferior to the full modalities used in contrastive learning. Actually, the reconstructed videos are visually worse than the full model (See Supplementary). Thus, it shows that the full progressive learning pipeline is crucial for the fMRI encoder to learn useful representations for this task. We also assess the reconstruction results without adversarial guidance. As a result, both numeric and visual evaluations decrease substantially. In fact, the generated videos can be highly similar sometimes, indicating that the negative guidance is critical in increasing the distinguishability of fMRI embeddings. See the Supplementary for more ablation studies and visual results.

## 5.1    Interpretation Results

We summarize the attention analysis in Fig. 7. We present the sum of the normalized attention within Yeo17 networks [23] in the bar charts. Voxel-wise attention value is displayed on a brain flat map, where we see comprehensive structural attention throughout the whole region. The average attention across all testing samples and attention heads is computed, revealing three insights into how transformers decode fMRI data.

**Dominance of visual cortex:** The visual cortex emerges as the most influential region. This region, encompassing both the central (VisCent) and peripheral visual (VisPeri) fields, consistently attracts the highest attention across different layers and training stages (shown in Fig. 7B). In all cases, the visual cortex is always the top predictor, which aligns with prior research, emphasizing the vital role of the visual cortex in processing visual spatiotemporal information[38]. However, the visual cortex is not the sole determinant of vision. Higher cognitive networks, such as the dorsal attention network (DorsAttn) involved in voluntary visuospatial attention control [39], and the default mode network (Default) associated with thoughts and recollations [40], also contribute to visual perceptions process [41] as shown in Fig. 7.

**Layer-dependent hierarchy:** The layers of the fMRI encoder function in a hierarchical manner, as shown in Fig. 7C, D and E. In the early layers of the network (panel A & C), we observe a focus on the structural information of the input data, marked by a clear segmentation of different brain regions by attention values, aligning with the visual processing hierarchy [42]. As the network dives into deeper layers (panel D & E), the learned information becomes more dispersed. The distinction between regions diminishes, indicating a shift toward learning more holistic and abstract visual features in deeper layers.

**Learning semantics progressively:** To illustrate the learning progress of the fMRI encoder, we analyze the first-layer attention after all learning stages, as shown in panel B: before contrastive learning, after contrastive learning, and after co-training with the video generation model. We observe an increase in attention in higher cognitive networks and a decrease in the visual cortex as learning progresses. This indicates the encoder assimilates more semantic information as it evolves through each learning stage, improving the learning of cognitive-related features in the early layers.

## 6    Conclusion and Discussion

**Conclusion** We propose MinD-Video, which reconstructs high-quality videos with arbitrary frame rates from brain activities represented by fMRI signals. Starting from large-scale pre-training to multimodal

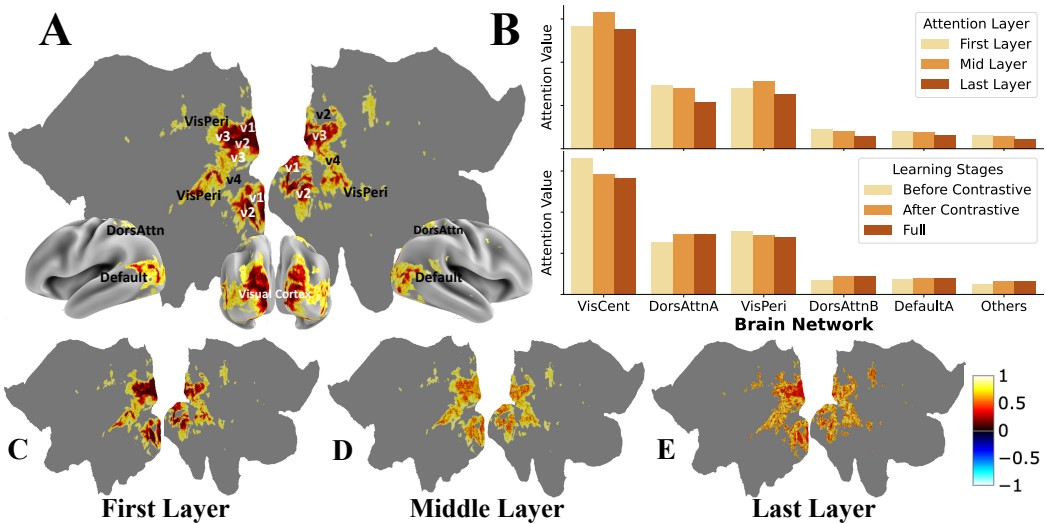

Figure 7. **Attention visualization**. We visualized the attention maps for different transformer layers (C, D, E) and learning stages (B) with bar charts and brain flat maps. Brain networks are marked on a brain surface map (A). Normalized attention values are shown.

contrastive learning with augmented spatiotemporal attention, our fMRI encoder learns features from a sliding time window of fMRI progressively. Then, we finetune an augmented stable diffusion for video generations, which is then co-trained with the fMRI encoder. Finally, we show that with fMRI adversarial guidance, MinD-Video recovers videos with accurate semantics, motions, and scene dynamics compared with the groundtruth, establishing a new state-of-the-art in this domain. We also show from attention maps that the trained model decodes fMRI with reliable biological principles.

**Limitations** Even though our method reconstructs videos with matching semantics and scenes, the pixel-level accuracy is not satisfactory, as shown in visual samples (more in the Supplementary). We believe that the results can be improved by introducing pixel-level information to control the diffusion denoising process, in addition to the visual semantic information decoded from the brain. Multi-scale information from the brain recordings can be fused to provide a more detailed visual description for generations.

Another limitation lies in the inter-subject generalization ability. Our method is still within the intra-subject level, and the inter-subject generalization ability remains unexplored due to individual variations. Additionally, our method only uses less than 10% of the voxels from the cortex for reconstructions, while using the whole brain data remains unexploited.

**Broader Impacts** We believe that brain decoding has promising applications in brain-computer interfaces as large models develop, and it will be a non-negligible component of the study to understand our cognitive process. However, governmental regulations and efforts from research communities are required to ensure the privacy of one's biological data and avoid any malicious usage of this technology.

# 7 Acknowledgements

This work was supported in part by National Medical Research Council, Singapore (NMRC/OFLCG19May-0035 to J-H Zhou), RIE2020 AME Programmatic Grant from A*STAR (to J-H Zhou), Ministry of Education Tier 2 Grant and Yong Loo Lin School of Medicine Research Core Funding (to J-H Zhou), National University of Singapore, Singapore.

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

# Supplementary Material

## A  Q&A Session

**Where does the performance gain come from?  Is it driven by the diffusion model only?** The performance gain in our model is not solely driven by the diffusion model. It is, in fact, a result of the combined efforts of two major stages: the fMRI encoder and the stable diffusion model.

In the first stage, the fMRI encoder plays a crucial role in learning representations from the brain. It effectively captures the complex spatiotemporal information embedded in the fMRI data, allowing the model to understand and interpret the underlying neural activities. This step is particularly important as it forms the foundation of our model and significantly influences the subsequent steps.

In the second stage, the stable diffusion model steps in to generate videos. One of the key advantages of our stable diffusion model over other generative models, such as GANs, lies in its ability to produce higher-quality videos. It leverages the representations learned by the fMRI encoder and utilizes its unique diffusion process to generate videos that are not only of superior quality but also better align with the original neural activities.

**What does the fMRI encoder learn? Why don't we train an fMRI to object classifier, followed by a generation model?** The fMRI encoder is designed to learn intricate representations from brain activity. These representations go beyond simple categorical information to encompass more nuanced semantic details that can't be adequately captured by discrete class labels (e.g. image texture, depth, etc.). Due to the richness and diversity of human thought and perception, a model that can handle continuous semantics, rather than discrete ones, is necessary.

The proposal to train an fMRI-to-object classifier followed by a generation model does not align with our goal of comprehensive brain decoding. This is largely due to a crucial trade-off between classification complexity and solution space:

- **Classification Complexity:** Classifying fMRI data into a large number of classes (e.g., 1000 classes) is non-trivial. As reported in [2], reasonable performance can only be achieved in a smaller classification task (less than 50-way), due to the limited data per category and the complexity of the task.

- **Limited Solution Space:** The solution space of discrete classes is significantly more restricted than that of continuous semantics. Thus, a classifier may not capture the complex, multi-faceted nature of brain activities.

This trade-off illustrates why a classifier might not be the best approach for this task. In contrast, our proposed method focuses on learning continuous semantic representations from the brain, which better reflects the complexity and diversity of neural processes. This approach not only improves the quality of the generated videos but also provides more meaningful and interpretable insights into brain decoding.

**Is the fMRI-video generation pipeline simply imputing missing frames in a sequence of static images based on the fMRI-image generation pipeline?** No, the fMRI-to-video generation process is not merely an imputation on the fMRI-to-image generation pipeline. While both involve generating visual content based on fMRI data, the tasks and their complexities are fundamentally different.

The fMRI-to-image generation involves mapping brain activity to a static image. This task primarily focuses on spatial information, that is, which brain regions are active and how they relate to elements in an image.

In contrast, the fMRI-to-video generation task involves mapping brain activity to dynamic videos. This task is considerably more complex as it requires the model to capture both spatial information and temporal dynamics. It's not just about predicting which brain regions are active, but also about understanding how these activations change over time and how they relate to moving elements in a video.

Adding to the complexity is the hemodynamic response inherent in fMRI data, which introduces a delay and blur in the timing of neural activity. This necessitates careful handling of the temporal aspects in the data. Furthermore, the temporal resolution of fMRI is quite low, making it challenging to capture fast-paced changes in neural activity.

We also use a stable diffusion process as our generative model, which is a probabilistic model. This means that the generation process involves a degree of randomness, leading to slight differences during each generation for video frames. Additionally, in video generation, we need to ensure consistency across video frames, which adds another layer of complexity.

**What's the major difference between Mind-Video and Mind-Vis [7]?** Our work extends beyond Mind-Vis by tackling the fMRI-based video reconstruction problem. Unlike image reconstruction, this adds another level of complexity. The key differences can be summarized in the following.

- **Problem formulation:** In Mind-Vis, dynamic fMRI recordings are averaged to create a "snapshot". While in this work, dynamic fMRI time series is directly used to recover a video, which requires considering the spatial features and the temporal features of fMRI. Additionally, the hemodynamic response is a significant challenge in our work, making the one-to-one mapping between fMRI and video even more difficult.

- **Architecture:** To address the unique challenges, we made two key improvements. First, we enhanced the fMRI encoder to handle a sliding time window of fMRI, capturing spatial and temporal information with distinct attention heads. Second, we employed multimodal contrastive learning to align fMRI with the semantic space of text and images before the co-training. This contrasts Mind-Vis in which co-training was performed directly without contrastive learning.

**Do all trainable models require re-training for each subject? How computationally demanding is it? How is the training process repeated for each subject independently?** Not all models necessitate re-training for every individual. The most resource-intensive phase is the large-scale pretraining of the MBM encoder, which remains consistent across subjects and does not demand re-training. Nevertheless, we do finetune both the MBM Encoder and the generative model using the fMRI data specific to each subject. This fine-tuning step, however, is relatively lightweight in terms of computational demands.

## B  More Implementation Details

**Large-Scale Pre-training** The large-scale pre-training uses the same setup as the MBM described in [7]. A ViT-large-based model with a 1-dimensional patchifier is trained with hyperparameters shown in Tab. B.1. The training takes around 3 days using 8RTX3090 GPUs. The training is performed on the 600,000 fMRI from HCP. Same as the literature, after the large-scale pre-training, the autoencoder is tuned with fMRI data from the target dataset, namely, Wen (2018), using MBM as well. The tuning is performed using a small learning rate and epochs.

| parameter | value | parameter | value | parameter | value | parameter | value |
|---|---|---|---|---|---|---|---|
| patch size | 16 | encoder depth | 24 | decoder embed dim | 512 | clip gradient | 0.8 |
| embedding dim | 1024 | encoder heads | 16 | learning rate | 2.5e-4 | weight decay | 0.05 |
| mask ratio | 0.75 | decoder depth | 8 | warm-up epochs | 40 | batch size | 500 |
| mlp ratio | 1.0 | decoder heads | 16 | epochs | 500 | optimizer | AdamW |

Table B.1. Hyperparameters used in the large-scale pre-training

**Multimodal Contrastive Learning** In this step, we will take the pre-trained fMRI encoder from the previous step and augment it with temporal attention heads to accommodate multiple fMRI. Then contrastive learning is performed with fMRI-image-text triplets. The image is a randomly-picked frame from an fMRI scan window. As mentioned, there are two important factors in the contrastive: batch size and data augmentations. Therefore, data augmentations are applied for all modalities. Random sparsification is used for fMRI, where 20% of voxels are randomly set to zeros each time. The random crop is applied to videos with a probability of 0.5. To augment the frame captions, we apply synonym augmentation and random word swapping. Due to a small dataset size ($\sim$4000), we use a dropout rate of 0.6 to avoid overfitting. For Subject 1 and 2, the training is performed with a batch size of 20, while a batch size of 32 is used due to fewer fMRI voxels with Subject 3. Training for all subjects is performed for 1,000 steps with a learning rate of $2 \times 10^{-5}$. The training takes around 10 hours using one RTX3090.

**Training of Augmented Stable Diffusion** The stable diffusion model is augmented with temporal attention heads for video generation. We train the augmented stable diffusion model with videos from the target datasets. The videos are downsampled from 30 FPS to 3 FPS at a resolution of $256 \times 256$ due to limited

GPU memory, even though our model can work with the full time resolution. This step is important for two reasons: 1) the augmented temporal heads are untrained; 2) the stable diffusion is pre-trained at a resolution of $512\times512$, so we need to adapt it to a lower resolution.

During the training, we update the self-attention heads ("attn1"), cross-attention heads ("attn2"), and temporal attention heads ("attn_temp"). The training is performed with text conditioning for 800 steps. We use a learning rate $2\times10^{-5}$ and a batch size of 14. The training takes around 2 hours using one RTX3090. Visual results show that videos of high quality can be generated with text conditioning after this step.

**Co-training** The fMRI encoder produces embeddings of dimensions $77\times768$, which are used to condition the augmented stable model during co-training. The whole fMRI encoder is updated, and only part of the stable diffusion is updated (same as the last step). The training is performed with a batch size of 9 and a learning rate of $3\times10^{-5}$ for 1,5000 steps. The training takes around 16 hours using one RTX3090.

**Inference** All samples are generated with 200 diffusion steps using fMRI adversarial guidance. The fMRI adversarial guidance uses an average fMRI as the negative guidance with a guidance scale of 12.5.

## C  Analysis of Visual Results

We test on all three subjects in Wen (2018) dataset. Around 6000 voxels are identified as ROI for Subject 1 and 2, while around 3000 voxels are identified for Subject 3. Thus, a larger batch size can be used when training with Subject 3, which may be the reason for its better numeric evaluation results. Nonetheless, all three subjects show consistent generation results. Some samples are shown in Fig. C.1.

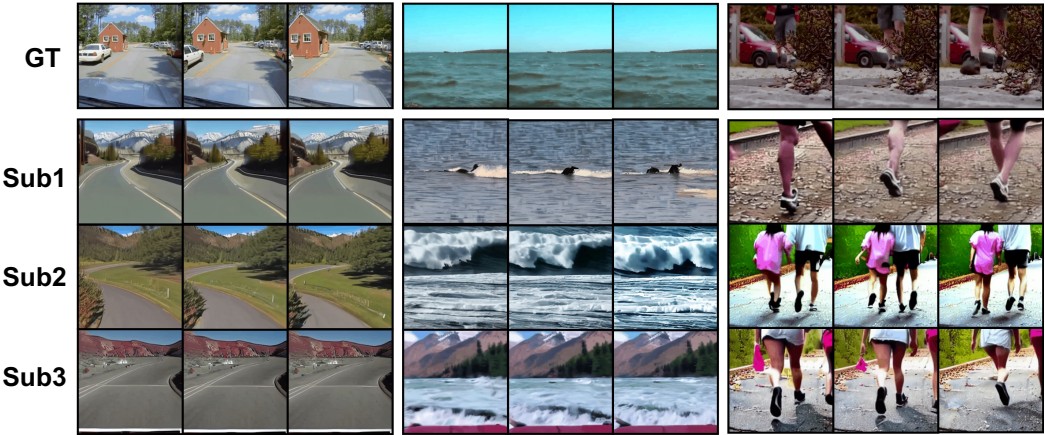

Figure C.1. Samples from different subjects.

Visual results of the ablation studies are shown in Fig. C.2. The Full model is trained with our full pipeline and inference with adversarial guidance. In contrastive learning ablation, we tested with incomplete modality, namely, image-fMRI and text-fMRI, respectively. Similar to the numeric evaluations, using incomplete contrastive gave an unsatisfactory visual result compared to using all three modalities. However, incomplete modality still outperformed inferencing without adversarial guidance significantly, which generated visually meaningless results.

Some fail cases are shown in Fig. C.3. It is observed that even though some fail cases generated different animals and objects compared to the groundtruth, other semantics like the motions, color, and scene dynamics can still be correctly reconstructed. For example, although the airplane and flying bird are not reconstructed, similar fast-motion scenes are recovered in Fig. C.3.

## D  More Ablation Studies

Other than the ablation studies shown in Tab. 1, we include extra ablation studies in Tab. D.2 to demonstrate the effectiveness of masked brain modeling (MBM) in our framework. Two experiments are performed.

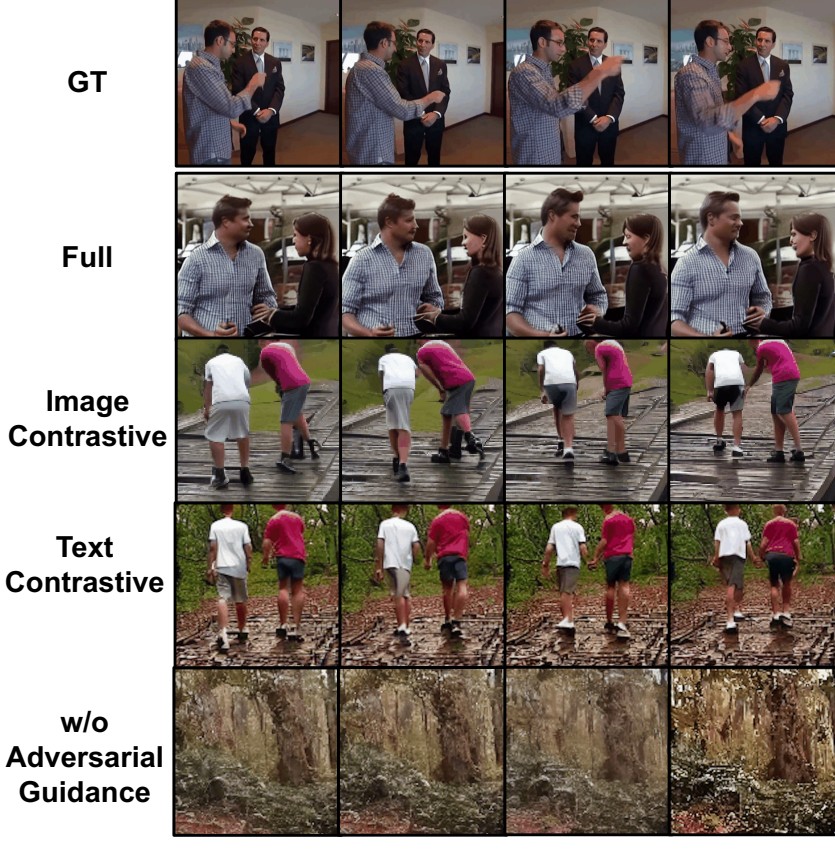

Figure C.2. Reconstruction samples for ablation studies. The Full model uses full modality contrastive learning with adversarial guidance.

The first experiment involves excluding the MBM pre-training, while the second experiment removes both the MBM pre-training and the contrastive training from the proposed method.

The results show the importance of MBM and contrastive training for all metrics. The performance drops as much as 55% without both components and 30% without only MBM. The visual quality of the generated videos also follows a similar trend as in Fig. C.2.

|  | 50-Way, Top-1 Accuracy | 50-Way, Top-1 Accuracy |  |
| --- | --- | --- | --- |
|  | Image Identification Tests | Video Identification Tests | SSIM |
| Full Model | 0.172±0.01 | 0.202±0.02 | 0.171 |
| w/o MBM, w/ Contrastive | 0.122±0.012 | 0.169±0.015 | 0.143 |
| w/o MBM, w/o Contrastive | 0.076±0.008 | 0.138±0.013 | 0.123 |

Table D.2. Extra ablation studies on masked brain modeling.

## E    More Evaluation Results

Limited by the publicly available video samples and codes for reproduction in the literature, we had some difficulties in comparing numerically with other methods. In the previous literature, Kupershmidt, 2022 [13] and Wen, 2018 [11] have only released part of their reconstructed videos, while Wang, 2022 [12] have not released their reconstructed videos. Also, because their released videos are based on different frame rates from ours (1Hz in [13], 0.5Hz in [11], 3Hz in ours), we did not perform the identification tests on those videos, as it might not be a fair comparison. Instead, we relied on a commonly available

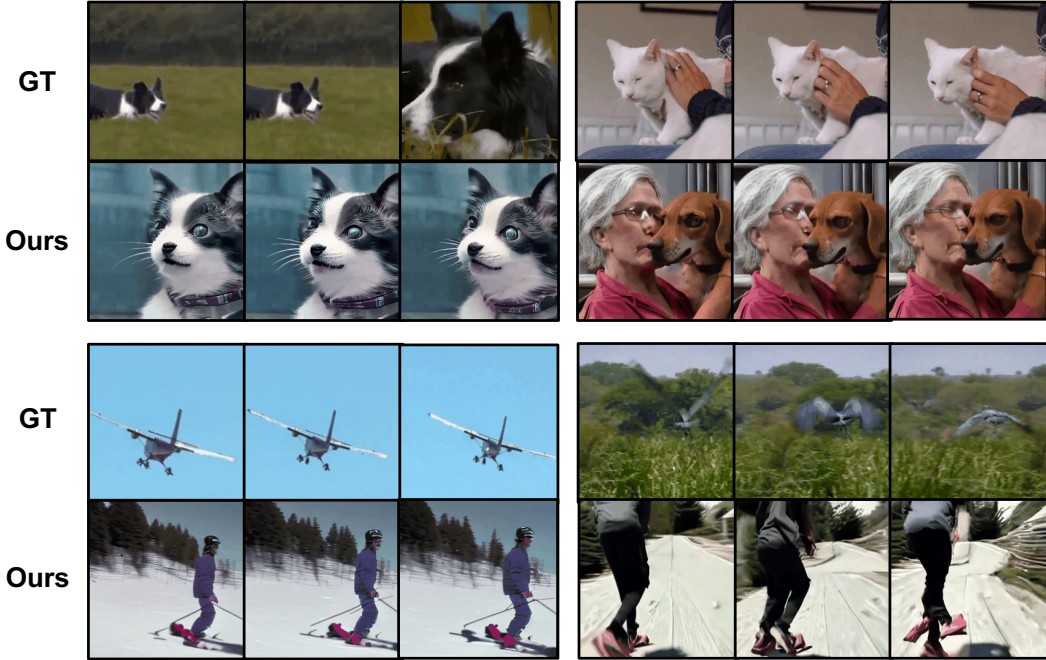

Figure C.3. Fail cases.

metric reported in their papers, SSIM. This metric was calculated quantitatively across all testing samples according to their papers.

Nevertheless, in this section, we perform the identification test on those incomplete samples only (released by previous work) and compare the results across different methods (including ours). Please see Tab. E.3. Our method outperformed the two previous methods in identification test accuracy.

| | 50-Way, Top-1 Accuracy | 50-Way, Top-1 Accuracy |
|---|---|---|
| | Image Identification Tests | Video Identification Tests |
| Ours | 0.195±0.016 | 0.265±0.02 |
| Kupershmidt, 2022 | 0.179±0.017 | 0.238±0.02 |
| Wen, 2018 | 0.07±0.01 | 0.166±0.016 |

Table E.3. Comparision of identification Tests with previous works.

## F  Source of Errors

In short, the failure cases can be attributed by two factors:

- Lack of pixel-level controllability. Due to the probabilistic nature of the diffusion model and the current conditioning method, the generation process lacks strong control from the fMRI latent to generate strictly matching low-level features, such as shapes, color, and geometric information. We believe this would be an important perspective for future research on this task.

- Uncontrollable factors during the scan. Mind wandering and imagination of the subject are usually inevitable during the scan. It has been shown that imagination is involved and can be decoded to some extent from the visual cortex, which can lead to mismatching between the ground truth and the generation results.

