# OpenReview forum: "Cinematic Mindscapes: High-quality Video Reconstruction from Brain Activity"
_NeurIPS.cc/2023/Conference — NeurIPS 2023 oral_

### Official Review · Reviewer_1EQ5 · 2023-06-16

**Soundness:** 3 good
**Presentation:** 3 good
**Contribution:** 2 fair
**Rating:** 6
**Confidence:** 4

**Summary:**

This paper aims to reconstruct high-quality video from brain activity. A novel model called MinD-Video is proposed, which could learn spatiotemporal information from continuous fMRI data.

**Strengths:**

1. It is quite interesting to reconstruct videos according to human brain regions' actions, Although the task is hard.
2. I believe all the newest methods are worth being tried to achieve fMRI data reconstruction, including Stable Diffusion and its video version.
3. The latent alignment and spatiotemporal attention are well-designed in this paper.

**Weaknesses:**

1. In Figure 4, we can see that Wen (2018) could reconstruct the shape and Kupershmit (2022) could reconstruct the texture. While videos generated by MinD-Video lack shape and texture. So I think previous works could affect more on the Video Generation part of MinD-Video.
2. The quantitative results are not sufficient. The metric values are not compared with other methods, and the gap between some experimental settings is limited.

**Questions:**

1. Please address my above concerns.
2. Do you think the Video Generation part should have controllability? I mean the current Video Generation part has no ability to reconstruct a video that is the same as source video, which would be the upper bound of the overall performance.

---

> ### Author Rebuttal · Authors · 2023-08-08
>
> We would like to thank the reviewer for the recognition of our contribution and the invaluable and constructive comments. Our point-by-point response is provided as follows.
>
> > 1. In Figure 4, we can see that Wen (2018) could reconstruct the shape and Kupershmit (2022) could reconstruct the texture. While videos generated by MinD-Video lack shape and texture. So I think previous works could affect more on the Video Generation part of MinD-Video.
>
> **Response:** Thank you for this insightful comment. Pixel level and semantic level decodings recover visual stimuli from two different perspectives, where the trade-off between fidelity and meaningfulness needs to be considered. In this work, **we prioritize the recovery of visual semantics** in fMRI, which is crucial for understanding the complex mechanism of human perception.
>
> **We recognize the significance of pixel and texture level information**, which contributes to generating visually closer results to the ground truth. For instance, leveraging results from previous methods could serve as a valuable guidance or initial starting point for our generative process.
>
> Nonetheless, our aim is to establish a foundation for future research that integrates both pixel-level features and visual semantics in this particular task. By doing so, we anticipate that future studies can further refine and enhance the decoding process, leading to even more comprehensive and accurate outcomes.
>
>
> > 2. The quantitative results are not sufficient. The metric values are not compared with other methods, and the gap between some experimental settings is limited.
>
> **Response:** We thank the reviewer for the constructive suggestion. **Limited by the publicly available video samples and codes** for reproduction in the literature, we had some difficulties in comparing numerically with other methods. In the previous literature, Kupershmidt, 2022 [12] and Wen, 2018 [10] **have only released part of their reconstructed videos**. Also, because their released videos are **basead on different frame rates from ours** (1Hz in [12], 0.5Hz in [10], 3Hz in ours), we did not perform the identification tests on those videos, which might not be a fair comparison. Instead, we relied on **a commonly available metric reported in their papers**, SSIM.  This metric was calculated quantitatively **across all testing samples** according to their papers. We did the same for our approach in the original paper.
>
> Nevertheless, as recommended, we have now decided to perform the identification test on those **incomplete samples** only (released by previous work) and compare the results across different methods (including ours). Please see the table below. Our method outperformed the two previous methods in identification test accuracy. We will include these results and  **possible limitations in the revised manuscript**.
>
> |                                            | 50-Way, Top-1 Accuracy     | 50-Way, Top-1 Accuracy   |
> |------------------------------------ |-------------------------------------|------------------------------------|
> |                                            | Image Identification Tests    | Video Identification Tests |
> | Ours                                   | **0.195 +- 0.016**                | **0.265 +- 0.02**              |
> | Kupershmidt, 2022             | 0.179 +- 0.017                     | 0.238 +- 0.02                    |
> | Wen, 2018                          | 0.07 +- 0.01                         | 0.166 +- 0.016                  |
>
> In order to strengthen our experimental settings, we performed extra ablation studies on different components of our designs on top of the existing ablation experiments. The first experiment involves **excluding the MBM pre-training**, while the second experiment **removes both the MBM pre-training and the contrastive training** from the proposed method. As shown in the table below, both the MBM pre-training and the contrastive training are critical to getting the best results, and removing any of them will incur a significant drop in performance.
>
>
>
> |                                            | 50-Way, Top-1 Accuracy     | 50-Way, Top-1 Accuracy   |          |
> |-------------------------------------|-------------------------------------|------------------------------------|---------|
> |                                            | Image Identification Tests    | Video Identification Tests | SSIM |
> | Full Model                          | **0.172 +- 0.01**                  | **0.202 +- 0.02**              | **0.171** |
> | w/o MBM, w/ Contrastive   | 0.122 +- 0.012                     | 0.169 +- 0.015                  | 0.143 |
> | w/o MBM, w/o Contrastive | 0.076 +- 0.008                     | 0.138 +- 0.013                  | 0.123 |
>
> > 3. Do you think the Video Generation part should have controllability? I mean the current Video Generation part has no ability to reconstruct a video that is the same as source video, which would be the upper bound of the overall performance.
>
> **Response:**  Thank you for your useful comment. Indeed, we agree that incorporating some form of controllability in the Video Generation part of our model could enhance the reconstruction of low-level image features like shape, texture, and location, bringing us closer to the source video. In fact, we believe that **controllability is a critical feature** for this kind of research, which is worth further research. We will include this part as future work in our revision.

---

> > ### Comment · Reviewer_1EQ5 · 2023-08-13
> >
> > Thanks for the author's elaborate response, and all my concerns have been well addressed. So I raise my rating as "Weak Accept".

---

> > > ### Author Response · Authors · 2023-08-14
> > >
> > > Many thanks for your support! We truly appreciate your precious time and valuable suggestions.

---

### Official Review · Reviewer_tnAn · 2023-07-05

**Soundness:** 2 fair
**Presentation:** 1 poor
**Contribution:** 1 poor
**Rating:** 3
**Confidence:** 5

**Summary:**

This paper focuses on the task of reconstructing human vision from brain activities. The authors propose MinD-Video that learns spatiotemporal information from continuous fMRI data of the cerebral cortex progressively through masked brain modeling, multimodal contrastive learning with spatiotemporal attention, and co-training with an augmented Stable Diffusion model that incorporates network temporal inflation. And the results are evaluated with semantic and pixel metrics at video and frame levels.

**Strengths:**

1. The authors provide both quantitative and qualitative results, and also provide some interpretable visualization results for demonstration.
2. The proposed method seems to perform better than previous non-diffusion methods.

**Weaknesses:**

1. The pre-training of fMRI encoder is composed of generative and contrastive objectives, which is very similar to previous works[1] that use masked contrastive pre-training in diffusion models. And the overall framework is a combination of existing methods or tricks (masked contrastive pretraining[1]+spatio-temporal attention (ST-Attn) mechanism[2]+stable diffusion[3]). Thus the novelty of this work is limited with respect to diffusion model field.

2. The proposed method contains two many tricks and modules, and causes no central point with respect to its technical contributions, the overall presentation is hard to follow.

3. I have reproduced the method, the results seem not as good as the demonstration in the paper and there sometimes exists a style inconsistency between GTs and generated results. The authors seem to cherry-pick the generated videos.

[1] Yang L, Huang Z, Song Y, et al. Diffusion-based scene graph to image generation with masked contrastive pre-training[J]. arXiv preprint arXiv:2211.11138, 2022.

[2] Wu J Z, Ge Y, Wang X, et al. Tune-a-video: One-shot tuning of image diffusion models for text-to-video generation[J]. arXiv preprint arXiv:2212.11565, 2022.

[3] Rombach R, Blattmann A, Lorenz D, et al. High-resolution image synthesis with latent diffusion models[C]//Proceedings of the IEEE/CVF conference on computer vision and pattern recognition. 2022: 10684-10695.

**Questions:**

1. The proposed method is of limited novelty, the authors need to rethink their main contributions.
2. The writing needs to be improved, and the central point needs to be emphasized.

**Limitations:**

See weakness part.

---

> ### Author Rebuttal · Authors · 2023-08-08
>
> We thank the reviewer for your time and effort in reviewing our work. We also appreciate your interest in our work and the useful suggestions. Our point-by-point responses are as follows.
>
> > 1. the novelty of this work is limited with respect to diffusion model field.
> > 2. The proposed method is of limited novelty.
>
> **Response:** We would like to take this opportunity to emphasize that though our method builds upon the established techniques, the novelty of our work is not in implementing these techniques per se. Instead, it lies in applying these techniques in a new and challenging domain: learning dynamic brain activities and reconstructing visual stimuli from the brain. Our work is at the intersection of neuroscience and CV, where the focus is not solely on inventing new tricks or structures for the diffusion model. Instead, **the main aim is to tackle the unique challenges of fMRI** and make proper methodological adjustments to adapt state-of-the-art generative models to our specific task. This has been recognized by the other four reviewers.
>
> Thanks for providing the additional reference. However, [1] provided a learning method for **scene graphs**, which is **an entirely different modality** from fMRI or any other brain recordings. Even though our method shares a similar philosophy with [1], this does not negate the novelty of our work. **The specific way of problem formulation, data modeling, and problem-solving is also an essential part of research novelty. Besides, the masking, contrastive, and generative objectives are standard techniques in representation learning.**
>
> Furthermore, we would like to highlight a few key differences.
> - We learn features of **dynamic fMRI with masking**, while [1] learns a **static scene graph without masking**.
> - Our work focuses on learning the **biological features** of fMRI, whereas [1] aims to learn the **geometric information** in the scene graph.
> - We must consider the **hemodynamic response**, a unique challenge when matching dynamic fMRI to videos. In contrast, the scene graph in [1] is static and matched to a static image.
> - We used a **3-modality** contrast among fMRI, video, and text to create a shared multi-modality space. While [1] used a **2-modality** contrast between scene graphs and images to discriminate a binary objective.
>
> In summary, while our method shares high-level similarities with [1] in terms of using generative and contrastive objectives, the comparison should not overlook the **different modalities and unique challenges** each work addresses. [1] deals with scene graphs, a static graphical structure, using an off-the-shelf graph encoder. Our work handles fMRI, a dynamic and biologically complex modality. Our problems involve understanding dynamic data subject to hemodynamic responses and the biological interplay of various brain regions - a quite different beast.
>
> We hope our work can encourage more research in this field and lead to more novel development in methods and applications. We are confident that with further reflection, the value of our approach will become more apparent.
>
> > 3. two many tricks. no central point
>
> **Response:** We apologize for the confusion. To clarify, all our modules and tricks are **summarized and depicted graphically** in Figure 2. Critical modules and methods are **color-coded and represented by different shapes** in Figure 2. Our technical contributions are summarized on Page 2 of our paper with **5 bullet points**, each containing two short sentences. As suggested, we will revise the paper to make our contributions and method descriptions clearer to improve readability.
>
> > 4. I have reproduced the method, the results seem not as good
>
> **Response:** We thank the reviewer for strong interest in our work and his/her effort to try to reproduce our findings. We are thrilled to see the level of engagement our work has elicited. To clarify, we assure you that we have presented a representative range of qualitative results of the generated videos in our paper. More importantly, we also evaluated the performance based on quantitative metrics and compared with other methods as presented in the original paper. Given this doubt from the reviewer, we have now included full generation samples in the ”Official Comment”.
>
> It would be great to know if such quantitative evaluations have been reproduced using the same data following the same process. It is worth noting that differences in outcomes could be due to various factors, including whether we are working with full samples or subsamples, data processing approaches, parameter settings, or the specific pretrain checkpoints used.
> Our codes will be made publicly available upon publication to ensure a fair reproduction for the community.
>
> Regarding the style inconsistency, we suspect the reviewer's concern is related to the visually distinct outcomes of generated results for sample inputs at each sampling. We acknowledge that this is a typical characteristic of the probabilistic nature inherent in the diffusion model, which represents one of the limitations of the diffusion-based decoding approach. However, despite these variations, **the semantic contents and evaluation metrics of the generated results remain consistent, as demonstrated in our paper**.

---

> > ### Comment · Reviewer_tnAn · 2023-08-13
> >
> > Thanks for the clarification, and it has addressed all my concerns. I will raise my rating.

---

> > > ### Author Response · Authors · 2023-08-14
> > >
> > > Many thanks for your support! We truly appreciate your precious time and valuable suggestions.

---

### Official Review · Reviewer_ZvgZ · 2023-07-06

**Soundness:** 2 fair
**Presentation:** 3 good
**Contribution:** 3 good
**Rating:** 6
**Confidence:** 4

**Summary:**

The paper presents a pipeline to decode videos from fMRI brain activity data. The pipeline is divided into several consecutive steps. First, a Transformer-based autoencoder is trained on an unsupervised Masked Brain Modelling task to learn fMRI representations on a large corpus of fMRI data. The fMRI encoder is then augmented with a spatial and temporal attention modules to enable the processing of temporal frames. Second, the fMRI encoder is further trained on a contrastive alignment task where fMRI representations are pulled closer to CLIP-based image and text representations of the corresponding video frames. Finally, a Stable Diffusion UNet model is augmented with temporal attention to allow its conditioning on the latents of the two previous frames and is finetuned end-to-end with the fMRI encoder. Experiments on a public dataset of participants watching videos inside an fMRI scanner show better performance (SSIM) as compared to existing baselines. Ablation studies are used to evaluate the impact of windowing hyperparameters and of different design choices. Attention weights from different Transformer layers are projected on a cortical map and visualized to highlight correspondence with different brain networks.

**Strengths:**

Originality: The combination of self-supervised learning, contrastive alignment in image/text latent space and conditional video generation for fMRI-to-video decoding is novel.

Quality: Most claims are supported (see below for possible exception to this). Qualitative and quantitative results are convincing.

Clarity: The manuscript is overall clearly written and well organized.

Significance: As one of the first fMRI-to-video approaches, this work is likely to inspire other work in the brain decoding literature. The qualitative and quantitative results suggest this is a clear improvement over existing baselines.

**Weaknesses:**

Quality: The underlying core claim of the submission, i.e. that the dynamic information contained in brain activity data can be used to reconstruct videos, might not be fully supported in the experimental settings presented in the paper (see Question 4). Essentially, it is not clear to me that the presented results prove temporally evolving brain activity can be decoded into videos; rather, it seems to suggest existing video diffusion models can be conditioned on a temporally-fixed (i.e. coming from time t) brain-derived latent representation.

Clarity: I found the description of the inputs at the contrastive learning stage unclear (Q3).

**Questions:**

1. What is the impact of the masked modelling pretraining on downstream performance? Given the training budget required for this step of the pipeline, it might be important to know the order of performance improvement it can bring.
2. I do not fully understand the idea behind the "network inflation trick" mentioned at line 157. Am I correct in thinking that what is called a batch here is built by taking consecutive fMRI frames, rather than by randomly sampling fMRI frames across e.g. a recording?
3. At the contrastive learning stage, what are the inputs to the fMRI encoder? Is it multiple fMRI frames, in which case there is a sequence of latent predictions per video segment, or is it still single fMRI frames matched to a single image frame? (The answer to this question might influence the relevance of the next question.)
4. I understand from line 231 that the decoded videos are generated from a single fMRI frame (yielding 6 video frames at 3 fps). In this particular case, the diffusion model is used in a similar setting as an fMRI-to-image model, i.e. it is conditioned by a unique brain-derived latent collected at time t. The resulting video could therefore be seen as an "hallucination" of the diffusion model based on an initial frame, rather than a truly dynamic decoded video, i.e. where temporal changes in brain activity drive changes in video frames. I assume that this might change if longer videos were generated e.g. based on 2 consecutive TRs. Is that something you have tried and if so, do the resulting videos remain temporally consistent when the conditioning is updated to a new fMRI frame?
5. Results of Table 1 show that removing adversarial guidance has a very mild effect on performance, especially for the video-based semantic metric. However corresponding results in Figure C2 suggests removing adversarial guidance dramatically impacts decoding performance. Does this just happen to be an unrepresentative example?

**Limitations:**

Yes.

---

> ### Author Rebuttal · Authors · 2023-08-08
>
> We are very grateful for your appreciation of the novelty and potential impact of our work and your important suggestions.  Our point-by-point responses are as follows.
>
> > 1. What is the impact of the masked modelling pretraining?
>
> **Response:** We thank the reviewer for raising this important point. We have now conducted additional ablation studies to thoroughly examine the impact of Masked Brain Modeling (MBM) in our approach. Specifically, we performed two experiments:
>
> 1. We conducted an experiment where we **excluded the MBM pre-training** from our proposed model.
> 2. We performed another experiment where **both the MBM pre-training and contrastive training were excluded** from our proposed model.
>
> For evaluation, we employed 50-way, top-1 identification tests, as detailed in the paper, along with Structural Similarity Index (SSIM) metrics. The results are tabulated below.
>
> | | 50-Way, Top-1 Accuracy| 50-Way, Top-1 Accuracy   |  |
> |-|-|-|-|
> |  | Image Identification Tests    | Video Identification Tests | SSIM |
> | Full Model | **0.172 +- 0.01**                  | **0.202 +- 0.02** | **0.171**|
> | w/o MBM, w/ Contrastive   | 0.122 +- 0.012| 0.169 +- 0.015 | 0.143 |
> | w/o MBM, w/o Contrastive | 0.076 +- 0.008| 0.138 +- 0.013 | 0.123 |
>
> As you can see from the Table, we observed a significant drop in performance when excluding MBM from our model, and this drop further intensified when both MBM and contrastive learning were excluded. Moreover, the visual quality of the generated results aligned well with these quantitative metrics. We will include both in our revised submission.
>
> > 2. I do not fully understand the idea behind the "network inflation trick" mentioned at line 157.
>
> **Response:** The batch here consists of **consecutive fMRI frames from a sliding time window**. The network inflation trick is a technique that enables the model to process an extra dimension by rearranging the input data shapes. For example, the transformer designed in previous work [6] for image reconstructions can only take input with 3 dimensions: n, p, and b, where n is the batch size and the last two dimensions will be used in the attention calculation in the transformer. Now we want to add an extra time dimension without changing the structure of the transformer such that the pre-trained weights provided in [6] can still be used. So to handle an input of shape (n, w, p, b), where w is the extra time dimension (sliding time window size), we can just merge two dimensions of the input into either (nw, p, b) or (np, w, b). When the dimension becomes (nw, p, b), the attention layer in the transformer is learning spatial correlations of the input, thus called spatial attention. When the dimension becomes (np, w, b), it learns the temporal correlations, thus called temporal attention. Combining spatial attention and temporal attention enables our model to learn spatial and temporal information from consecutive fMRI frames.
>
>
> > 3. I found the description of the inputs at the contrastive learning stage unclear (Q3).
>
> **Response:** We apologize for the confusion. The input to the fMRI encoder at the contrastive learning stage is **a sliding time window of fMRI, i.e., multiple fMRI frames, which are matched to a video clip**. We will provide more details in our response to the next question. The output of the fMRI encoder is an embedding that is learned by considering both the spatial and the temporal features of the fMRI frames. We will revise our manuscript to make this point clearer.
>
>
> > 4. Question 4
>
> **Response:** We apologize again for the confusion. In line 231 of our paper, we are meant to say each fMRI frame corresponds to 2 seconds of video, as the TR of fMRI scanning is 2 seconds. However, considering the nature of hemodynamic response function of the human brain, a 2-second video may be encoded by fMRI frames at both t and t+1. Therefore, we create a sliding time window that groups multiple consecutive fMRI frames, which is used as input to the fMRI encoder. For example, consider a consecutive fMRI and videos represented by [f1, f2, f3, f4] and [v1, v2, v3, v4]. Assuming a window size of 2, we will use [(f1, f2, v1), (f2, f3, v2), (f3, f4, v3) … ] as the (fMRIs, video) pairs for training and testing. As we can see, the input to our pipeline is actually consecutive fMRI frames from a sliding time window. For consecutive fMRI-based sliding windows, the decoding results will be highly similar if they are matched to a similar groundtruth. However, there could be some style inconsistency between consecutive fMRI windows due to the probabilistic nature of the diffusion model, which we believe could be better improved in future research.
>
> Another clarification is that we only reconstructed a 2-second video each time because of memory limitation of our GPU. Longer video generation requires larger GPU memory. Nevertheless, thanks to the sliding time window design, we can theoretically expand our approach and reconstruct longer videos with a larger fMRI window size with enough GPU memory.
>
> > 5. Results of Table 1 show that removing adversarial guidance has a very mild effect.
>
> **Response:** To clarify, without adversarial guidance, most generated videos are visually worse than our full method. In Table 1, it does drop from 0.172 to 0.117 in frame-based semantic level metrics and from 0.171 to 0.152  in the SSIM. However, even though the results are visually worse, **some low-level features are still generated**, such as animal-like moving objects, blue burry water-like objects, etc, with matching motions, which can still be classified into a correct semantic category in the metrics. Overall, adversarial guidance is still helpful.

---

> > ### Comment · Reviewer_ZvgZ · 2023-08-13
> >
> > Thank you to the authors for the additional ablations and for the clarifications.
> >
> > Thank you also for providing the generated videos. I looked through many of them and while there are clear success cases where the generation looks a lot like the stimuli, there are also many examples where the generation appears unrelated to the ground truth category. In some rarer cases the generation even looks like noise or is devoid of any clear semantics. I am concerned that the current figures in the manuscript (Fig. 1, 4 and 5) along with the description of the results and the discussion do not clearly mention or show examples of these failure cases.
> >
> > As the approach is novel and the results offer a strong baseline for further work on this topic I keep my current score. However I believe the manuscript should reflect more transparently what a significant portion of the generated videos look like.

---

> > > ### Author Response · Authors · 2023-08-14
> > > **Response to the Revewer's Comment**
> > >
> > > Thanks for your reply! We agree and understand your concern. We discussed the failure cases in the supplementary material of our original submission. In the revision, we will include more failure cases and add a clear notation for the failure cases in the discussion of the results. We will also discuss the failure cases in the limitation section to give a clearer overview and explanation for the failure cases.
> > >
> > > In short, the failure cases can be attributed into two categories.
> > >
> > > 1. Lack of pixel-level controllability. Due to the probabilistic nature of the diffusion model and the current conditioning method, the generation process lacks strong control from the fMRI latent to generate strictly matching low-level features, such as shapes, color, and geometric information. We believe this would be an important perspective for future research on this task.
> > >
> > > 2. Uncontrollable factors during the scan. Mind wandering and imagination of the subject are usually inevitable during the scan. It has been shown that imagination is involved and can be decoded to some extent from the visual cortex [4], which can lead to mismatching between the ground truth and the generation results.
> > >
> > > We would also like to highlight that the quantitative evaluation across all samples reported in our original submission is compared with the reported results from the literature. We also compare with Kupershmidt, 2022 [12] and Wen, 2018 [10] using their released videos (partial testing set), as shown in the table below. We achieved better numeric results in these comparisons.
> > >
> > > Thanks again for the suggestions, which enhanced the quality of our manuscript! We hope this reply has addressed your concern.
> > >
> > >
> > > |                                            | 50-Way, Top-1 Accuracy     | 50-Way, Top-1 Accuracy   |
> > > |------------------------------------ |-------------------------------------|------------------------------------|
> > > |                                            | Image Identification Tests    | Video Identification Tests |
> > > | Ours                                   | **0.195 +- 0.016**                | **0.265 +- 0.02**              |
> > > | Kupershmidt, 2022             | 0.179 +- 0.017                     | 0.238 +- 0.02                    |
> > > | Wen, 2018                          | 0.07 +- 0.01                         | 0.166 +- 0.016                  |
> > >
> > > [4] G. Shen, T. Horikawa, K. Majima, and Y. Kamitani, “Deep image reconstruction from human brain activity,” PLoS computational biology, vol. 15, no. 1, p. e1006633, 2019
> > >
> > > [10] H. Wen, J. Shi, Y. Zhang, K.-H. Lu, J. Cao, and Z. Liu, “Neural encoding and decoding with deep learning for dynamic natural vision,” Cerebral cortex, vol. 28, no. 12, pp. 4136–4160, 2018
> > >
> > > [12] G. Kupershmidt, R. Beliy, G. Gaziv, and M. Irani, “A penny for your (visual) thoughts: Self-supervised reconstruction of natural movies from brain activity,” arXiv preprint arXiv:2206.03544, 2022

---

> > > > ### Comment · Reviewer_ZvgZ · 2023-08-15
> > > >
> > > > Thank you for the additional answer, this has addressed my remaining concern.

---

### Official Review · Reviewer_4NZX · 2023-07-10

**Soundness:** 3 good
**Presentation:** 3 good
**Contribution:** 3 good
**Rating:** 8
**Confidence:** 3

**Summary:**

This research proposes a method called Mind-Video to reconstruct videos from brain activity. By utilizing continuous fMRI data and advanced techniques, Mind-Video can generate high-quality videos with arbitrary frame rates. The model outperforms previous methods in accuracy and structural similarity.


**Strengths:**

The present work contributes with an innovative approach for reconstructing continuous visual experiences from brain activities. By utilizing masked brain modeling, multimodal contrastive learning with spatiotemporal attention, and co-training with an augmented Stable Diffusion model, their method, called Mind-Video, surpasses previous techniques in reconstructing high-quality videos.

This paper presents a meticulous study of previous work, which is important in the development of the present work. Also, the technical aspects are clearly explained and have also been evaluated using the correct metrics.

The experimental evaluation of the proposed model demonstrates its superior performance compared with other works.

The developed methodology provides interpretability, which is a very important factor in medical applications.

The interpretation of the results is good, which strengthens the results and the paper in general.

In summary, the work is an interesting application of deep learning in the medical area, and it also has a remarkable novelty.

**Weaknesses:**

The work is very interesting, the resulting video resembles the ground true in terms of activity, but in terms of scene there is still a considerable difference.

In the "Paired fMRI-Video dataset" part, only three subjects are used, this is a limitation in terms of generalization. It is a limitation but it would be interesting to use more subjects and have more generalizable results. Nevertheless, it is ahead of several previous works.

**Questions:**

None

**Limitations:**

The limitations are well explained by the authors.

---

> ### Author Rebuttal · Authors · 2023-08-08
>
> We thank you for the strong support and the positive comments on our work. Your inspiring questions and comments are valuable for our future work. Our point-by-point responses are as follows.
>
> > 1. The work is very interesting, the resulting video resembles the ground true in terms of activity, but in terms of scene there is still a considerable difference.
>
> **Response:**  We appreciate the useful feedback on the generated videos. We agree that while our model captures activity patterns well, scene reconstruction still presents challenges. This stems from the significantly **lower signal-to-noise ratio and increased complexities in the spatial-temporal dynamics** in the paired fMRI-video data compared to the paired fMRI-image data. Furthermore, there is higher inherent variability in **individuals' imaginations due to the dynamic nature of videos (in contrast to static images)**. In the proposed model, we focused more on the video semantic recovery than the low-level visual features. We are aligned with the reviewer that this limitation should be addressed in future research.
>
> > 2. In the "Paired fMRI-Video dataset" part, only three subjects are used, this is a limitation in terms of generalization. It is a limitation, but it would be interesting to use more subjects and have more generalizable results. Nevertheless, it is ahead of several previous works.
>
> **Response:** We appreciate the reviewer sharing his/her valuable perspective on this work.  **We totally agree that utilizing more subjects can help evaluate and potentially enhance the generalizability of our method.** Unfortunately, this is subject to the availability of suitable fMRI-video datasets currently. This is a common problem for this type of neuroscience application. We will acknowledge cross-subject model generalization as a crucial **future work direction** in the revised manuscript. More work is needed to perform well-designed brain recording experiments and methodological development to increase the generalizability and interpretability.

---

> > ### Comment · Reviewer_4NZX · 2023-08-21
> > **Final remarks**
> >
> > I appreciate your prompt and comprehensive response to my review of your paper. Your detailed answers have better explained various aspects of the paper, and I find them very useful to better understand the approach and direction of the paper.
> >
> > Your explanation of the challenges in scene reconstruction and activity pattern capture are insightful and aligns with my observations. I appreciate your acknowledgment of this limitation and your plans to address it in future research.
> >
> > In regard to the "Paired fMRI-Video dataset" limitation, I understand the constraints imposed by the availability of suitable datasets. Your willingness to acknowledge this limitation and incorporate the importance of model generalization in future research revisions is a step in the right direction. I agree that well-designed brain recording experiments and methodological development are necessary to enhance generalization and interpretability.
> >
> > Once again, I would like to express my appreciation for your responses to my comments. Your willingness to engage in discussions reflects your dedication to advancing your research and addressing the concerns of the reviewers. I am confident that your efforts will contribute positively to the field of deep learning in medical applications.

---

> > > ### Author Response · Authors · 2023-08-21
> > >
> > > Many thanks again for your support! We sincerely appreciate your valuable comments and your precious time and efforts in reviewing our paper!

---

### Official Review · Reviewer_19wY · 2023-07-25

**Soundness:** 3 good
**Presentation:** 4 excellent
**Contribution:** 3 good
**Rating:** 7
**Confidence:** 3

**Summary:**

The authors have developed an fMRI to video model trained using contrastive learning and stable diffusion. The generted videos are evaluated based on the semantics of their content and pixel level metrics at video and frame level, some of which utilize pretrained classifiers trained on ImageNet and VideoMAE. The work builds on top of and shares attributes with the MinD-Vis model including the MBM approach. It constitues of two modules: fMRI encoder and a video generative model, trained separately, and finetuned together.

**Strengths:**

The paper proposes a sound architecture constitued of Spatial and Temporal attention, multimodal contrastive learning, adversarial guidance, and diffusion models. The end2end pre-processing and training process, including usage of pretrained models such as BLIP for video captioning have a few introguing novelties.


**Weaknesses:**

Eventhough reference [6] is limited to generating images, the differentiating factors between the current work and reference [6] is better be highlighted in more details.

In the "Adversarial Guidanc for fMRI", there is a claim about using the "average all fMRI in the testing set as the negative guidance". Usage of the "testing" set in the training process, even in its aggregated (averaged) sense is not the best design.

Given the complexity of the architecture, the ablation study could have been improved to highlight the impact of more components.

Assuming the trainable modules require re-training per subject, the proposed design raises concerns around practicality of the approach.

**Questions:**

In the "Multimodal Contrastive Learning" subsection, the idea of {fMRI, video, caption} triplet needs further explanation. If embeddings of fMRI, video and caption encoded via three separate encoders, what does the "augmented endcoder" refer to in this context?

In the same subsection, the idea of concatenating captions with "Then" in-between is discussed. How impactful was the addition of "Then" keyword in-between the two captions? Did you run an ablation study that demonstrates positive effect?

In the "Scene-Dynamic Sparse Causal (SC) Attention" subsection, with regards to the attention maps, we wonder if the authors monitored attention-maps across transition frames; i.e., did they observe the attention to previous frames (i-2 and i-1) to disipate during transitions?

Do all trainable models (e.g. SC-MBM Encoder, etc.) require re-training for each subject?


**Limitations:**

Yes.
The intra-subject limitation of the method is highlighted; however, it is not clear whether the training process was repeated for each subject independently.

---

> ### Author Rebuttal · Authors · 2023-08-08
>
> We truly appreciate your recognition of our contributions and novelty.  Our point-by-point responses to the comments are as follows.
>
> > 1. The differentiating factors between the current work and [6] is better be highlighted.
>
> **Response:** We thank the reviewer for this important point. Our work extends beyond [6] by tackling the fMRI-based video reconstruction problem. Unlike image reconstruction, this adds another level of complexity. The key differences can be summarized in the following. We will revise our paper to highlight this point.
>
> - **Problem formulation**. In [6], dynamic fMRI recordings are averaged to create a “snapshot”. While in this work, **dynamic fMRI time-series** is directly used to recover a video, which requires considering the **spatial features** and the **temporal features** of fMRI. Additionally, the **hemodynamic response** is a significant challenge in our work, making the one-to-one mapping between fMRI and video even more difficult.
>
> - **Architecture**. To address the unique challenges, we made two key improvements. First, we enhanced the fMRI encoder to handle a sliding time window of fMRI, capturing spatial and temporal information with distinct attention heads. Second, we employed **multimodal contrastive learning** to align fMRI with the semantic space of text and images before the co-training. This contrasts [6] in which co-training was performed directly without contrastive learning.
>
>
> > 2. Usage of the "testing" set in the training process is not the best design.
>
> **Response:** To clarify, the adversarial guidance was only used in the **inference stage (testing)** to create a stronger condition and to increase the signal-to-noise ratio of the testing fMRI dataset. The averaged testing fMRI data was not involved in any part of the training process.
>
> > 3. Given the complexity of the architecture, the ablation study could have been improved.
>
> **Response:** As recommended, we have now conducted additional ablation studies to examine the impact of two crucial components in our approach: masked brain modeling (MBM) pre-training and contrastive training.
>
> We performed two new experiments on top of the existing ablation experiments in the table below. The first experiment involves **excluding the MBM pre-training**, while the second experiment **removes both the MBM pre-training and the contrastive training** from the proposed method.
>
> The results show the importance of MBM and contrastive training for all metrics. The performance drops as much as 55% without both components and 30% without only MBM. The visual quality of the generated videos also follows a similar trend, which will be detailed in our revised manuscript.
>
> Again, we sincerely appreciate your comment. These additional experiments strengthen the empirical evidence supporting our proposed approach.
>
> | | 50-Way, Top-1 Accuracy| 50-Way, Top-1 Accuracy| |
> |-|-|-|-|
> | | Image Identification Tests| Video Identification Tests | SSIM |
> | Full Model| **0.172 +- 0.01**| **0.202 +- 0.02**| **0.171**|
> | w/o MBM, w/ Contrastive   | 0.122 +- 0.012| 0.169 +- 0.015| 0.143 |
> | w/o MBM, w/o Contrastive | 0.076 +- 0.008| 0.138 +- 0.013| 0.123 |
>
> > 4.  Assuming the trainable modules require re-training per subject, the proposed design raises concerns around practicality.
>
> **Response:** We acknowledge the practical concern regarding per-subject re-training process. But it is worth noting that almost all the existing methods for this specific neuroscience application, i.e., “brain decoding”, rely on per-subject training due to high inter-subject variability and limited datasets. Cross-subject model generalization remains an **open problem** and an exciting **future work direction** of our group and others.
>
> > 5. What does the "augmented encoder" refer to in this context?
>
> **Response:** The fMRI encoder in [6] can only process a single fMRI frame without considering the temporal dynamics of the fMRI recordings (i.e., multiple frames). We changed its architecture to **encode multiple fMRI frames in a sliding time window manner**, considering **spatial and temporal correlations** during feature learning. Thus, it is called an “augmented encoder”.
>
> > 6. How impactful was the addition of "Then" keyword in-between the two captions?
>
> **Response**:  We apologize for the confusion. To clarify, "Then" concatenation was only used in the augmented stable diffusion training (not the contrastive learning). This stage does not impact the fMRI feature learning process significantly, thus having **minimal influence on the final results**. To improve clarity, we will relocate the captions of the concatenation part to the stable diffusion training subsection in our revision.
>
> > 7. We wonder if the authors monitored attention-maps across transition frames?
>
> **Response:** Yes, we did monitor attention maps across transition frames. However, we **did not observe** that the attention to the previous frames noticeably dissipated during the transitions, which led to frames containing parts of the content from the two scenes.
>
> This can be attributed to the nature of our generated videos: they span only 2 seconds, mirroring the 2-second TR of the fMRI. Given this brief duration, distinguishing between transition and non-transition frames becomes inherently challenging. However, the scene-changing decoding is an exciting question and an unexplored field worth further research.
>
> > 8. Do all trainable models require re-training for each subject?
> > 9. it is not clear whether the training process was repeated for each subject independently.
>
> **Response:** No. The most resource-consuming part is the large-scale pretraining of the MBM encoder, which **does not require re-training** for each subject. But we do **need to finetune** both the MBM Encoder and the generative model using each subject's fMRI. However, this process is **not computationally expensive**. We will make this part clearer in the revision.

---

> > ### Comment · Reviewer_19wY · 2023-08-18
> >
> > Thank you for the updates, responses and further evaluations. My current rating remains valid.

---

> > > ### Author Response · Authors · 2023-08-18
> > >
> > > Thanks again for your time and efforts. We really appreciate your support!

---

### Author Rebuttal · Authors · 2023-08-08

We are grateful to all five reviewers and AC/SACs for their valuable time, insightful comments, and useful suggestions. We will carefully revise our paper according to the comments. Our point-by-point response to the reviewers’ comments has been added to the individual chat box for each reviewer. We believe that the revised manuscript has been enhanced and the concerns have been well addressed.

Due to the character limitation, citations mentioned in the rebuttal are included below.

[1] Yang L, Huang Z, Song Y, et al. Diffusion-based scene graph to image generation with masked contrastive pre-training[J]. arXiv preprint arXiv:2211.11138, 2022.

[2] Wu J Z, Ge Y, Wang X, et al. Tune-a-video: One-shot tuning of image diffusion models for text-to-video generation[J]. arXiv preprint arXiv:2212.11565, 2022.

[3] Rombach R, Blattmann A, Lorenz D, et al. High-resolution image synthesis with latent diffusion models[C]//Proceedings of the IEEE/CVF conference on computer vision and pattern recognition. 2022: 10684-10695.

[6] Z. Chen, J. Qing, T. Xiang, W. L. Yue, and J. H. Zhou, “Seeing beyond the brain: Masked modeling conditioned diffusion model for human vision decoding,” in Proceedings of the IEEE/CVF Conference on Computer Vision and Pattern Recognition (CVPR), 2023.

---

### Author Response · Authors · 2023-08-10
**Link to Video Files**

Thanks for the time and efforts from the reviewers, ACs, and SACs again.

As requested by Reviewer tnAn, the link to our **full generated videos** is listed below. As Reviewer tnAn asked for full samples, the link is an anonymous link containing 1.6GB data (~3600 gif files).

https://figshare.com/s/5b596aeb6beebd32d2d1

Many thanks again!

---

### Decision · Program_Chairs · 2023-09-21

**Decision:**

Accept (oral)

**Comment:**

The paper has received positive feedback from the reviewers. There were some concerns that were addressed by the rebuttal. this paper presents a great contribution to the field of computational neuroscience. The authors are encouraged to address the comments in their final paper.